# Perceived barriers for accessing international research funding among Latin American researchers

Jesica Formoso[1,2,3]*, Paz Miguez[3,4], Nicolás Palopoli[3,5], Irene Vazano[3], Julián Buede[3], Juan Pablo Barreyro[1,2], Laura Ación[3,4]

**1** Centro Interdisciplinario de Investigaciones en Psicología Matemática y Experimental, Ciudad de Buenos Aires, Argentina, **2** National Scientific and Technical Research Council, Ciudad de Buenos Aires, Argentina, **3** MetaDocencia, Portland, United States of America, **4** Universidad de Buenos Aires, Ciudad de Buenos Aires, Argentina, **5** Universidad Nacional de Quilmes, Buenos Aires, Argentina

\* jformoso@conicet.gov.ar

## Abstract

Access to international funding is increasingly essential for sustaining research in a highly competitive and globalized scientific landscape; however, researchers in Latin America face persistent structural constraints that limit their competitiveness. This study aimed to identify perceived barriers and strategies to mitigate structural disadvantages in accessing international research funding. We employed a sequential mixed-methods design. In the qualitative phase, a virtual focus group with five researchers from Mexico, Colombia, Argentina, and Guatemala explored experiences with international grant applications. Thematic analysis revealed three overarching themes: (1) strategies and good practices; (2) barriers and structural constraints; and (3) recommendations to support more inclusive funding practices. Barriers included language and cultural differences, perceived bias from Global North funders, differences in infrastructural expectations, and limited opportunities for networking. These insights informed the design of a quantitative survey, which was completed by 253 researchers from 16 countries; 60.4% had applied to international calls and 43.6% reported receiving at least one grant. Across 668 reported applications, success rate was 40.6%. The most frequently endorsed barriers were economic costs of networking (92.3%), misalignment between eligibility criteria and local trajectories (80.8%), rhetorical differences in grant writing (75.3%), and self-perceptions of lower competitiveness (69.2%). Results highlight persistent structural, linguistic, and cultural barriers that shape access to international funding. Participants proposed actionable strategies—particularly for international funders seeking to broaden Global South participation—including context-sensitive grant-writing training, mentorship schemes, and funding calls explicitly designed to account for the Latin American realities.

**Data availability statement:** All data, materials, and analysis scripts are available in the Open Science Framework (OSF) repository at http://doi.org/10.17605/osf.io/uvdqf.

**Funding:** This work was supported by the Chan Zuckerberg Initiative DAF, an advised fund of the Silicon Valley Community Foundation (Grant #2024-349265 (5022) GB-1616778; https://chanzuckerberg.com/). The funders had no role in study design, data collection and analysis, decision to publish, or preparation of the manuscript.

**Competing interests:** The authors have declared that no competing interests exist.

## Introduction

Scientific research has become a highly globalized enterprise, increasingly organized through international collaboration networks, competitive funding schemes, and standardized evaluation criteria. While global integration has expanded opportunities for knowledge circulation, it has also reinforced long-standing structural imbalances in research capacity across countries and regions [1]. Researchers based in high-income countries typically have access to more robust funding, infrastructure, institutional backing, and international research networks, whereas researchers in low-to-middle-income countries (LMICs) encounter persistent structural constraints associated with comparatively fragile and under-resourced national scientific systems [2,3]. As the number of researchers competing for limited funding continues to grow faster than overall research budgets, these pre-existing differences increasingly translate into asymmetric research capacities and highly uneven scientific outputs [4].

These disparities are evident in global patterns of investment in research and development (R&D). On average, countries belonging to the Organisation for Economic Co-operation and Development (OECD) invest around 2% of their GDP in research and development, while many Latin American and African countries remain below 0.7% [5,6]. In Latin America, even though many countries have consolidated scientific systems supported by national agencies–such as the National Scientific and Technical Research Council (CONICET) in Argentina, the National Council of Science and Technology (CONACYT) in México, and the National Agency for Research and Development (ANID) in Chile—the predominance of public funding, together with these systems' vulnerability to economic cycles and political fluctuations, undermines the long-term sustainability of scientific agendas [7,8]. Public investment in research and development is primarily oriented toward advancing domestic priorities such as economic competitiveness, innovation, and social development, while international R&D funding that relies on public resources is often guided by strategic considerations rather than by redistributive objectives [9,10]. As a result, reliance on funders and donors from larger economies may limit the extent to which Latin American researchers and institutions are able to pursue research priorities closely aligned with local needs [11,12].

Importantly, the challenges faced by researchers in less well-resourced systems are not limited to financial constraints. Structural asymmetries are also embedded in the very systems that govern how research is funded and valued. These mechanisms operate at the intersection of multiple social and contextual factors, shaping researchers' capacity to participate in international competitions and networks. Language, for example, functions as a key gatekeeping mechanism in global science. English has consolidated its position as the dominant language of scientific communication and is often a requirement for visibility and recognition. Studies indicate that more than 90% of academic publications are written in English, that these communications tend to receive higher citation rates, and that the journals publishing primarily in English have higher impact than those in other languages [13–15]. For non-native English speakers, however, these conditions entail additional time, effort, and resources [16–18]. Nationality adds another layer, as researchers from the Global South, especially

those from underrepresented groups, often navigate visa restrictions, bureaucratic burdens, unfavorable exchange rates, and exclusionary practices that their peers in wealthier countries rarely face [19,20].

Furthermore, grant writing itself constitutes a specialized professional skill that requires deliberate training, familiarity with funders' expectations, and institutional support. In many high-income countries (HICs), researchers operate within well-established support structures—such as research development offices, grant administrators, internal review processes, mentorship, and targeted training—that enhance preparedness for highly competitive funding calls [21]. By contrast, researchers in less well-resourced scientific systems often have limited access to such support, with proposal development frequently absorbed into existing academic workloads.

Collectively, these structural and contextual factors shape not only career trajectories but also the broader participation by Latin American researchers in international scientific events and networks. The literature highlights that Latin American researchers often experience lower success rates in international grant competitions, identifying barriers such as complex application procedures, limited institutional support, and differences between donor priorities and local challenges [20]. These findings suggest that structural conditions within national research systems play a significant role in shaping researchers' competitiveness in international funding calls.Despite this growing body of work, much of the existing research on barriers to international research funding in the region relies on bibliometric analyses or opinion pieces [7,20,22]. Relatively few studies incorporate the direct perspectives of researchers themselves. One notable exception is the study by Huete-Pérez and Salvatierra [23], which used a mixed-methods design to assess biomedical research capacity in Colombia, Costa Rica, Guatemala, Panama, and Peru. Through expert interviews and surveys, the authors identified critical barriers, including insufficient national funding, dependency on external grants, weak postgraduate training opportunities, and fragile research infrastructures. This work underscores the value of combining qualitative insights with broader survey data to better understand how structural conditions shape researchers' experiences with international funding.

Building on this gap, the present study aims to identify the structural and institutional factors that shape Latin American researchers' ability to compete for international research funding. By examining researchers' experiences with international funding calls, the study seeks to generate evidence that can inform context-sensitive strategies and capacity-building initiatives aimed at strengthening researchers' competitiveness within global funding environments. To this end, we conducted a mixed-methods study to examine the barriers and challenges faced by Latin American researchers when applying for international research funding. The initial qualitative phase involved a focus group with researchers from multiple Latin American countries, specifically designed to analyze their narratives and experiences related to submitting proposals to international funding calls. Insights from this focus group directly informed the design of a survey, which was tailored to assess the prevalence and variability of perceived barriers to applying for and securing funding, and subsequently used in the quantitative phase of the study.

## Methods

### Participants

For the qualitative phase of this mixed-methods study, we invited five researchers from Mexico, Colombia, Argentina, and Guatemala, representing a range of academic career stages from postdoctoral to mid-career. Recruitment followed a snowball sampling strategy, a non-probabilistic method chosen to maximize diversity in geographic location, career stage, and research area. Eligibility criteria required that participants: (1) be fluent in Spanish; (2) be currently working in Latin America in a biomedical-related field; (3) have between three and twenty years of postdoctoral research experience; (4) demonstrate interest in fundraising, capacity building, and community development; and (5) possess documented experience in inclusive communities focused on strengthening research capacity. Recruitment for the qualitative phase took place between March 1, 2025 and April 10, 2025. All participants were adults and provided written informed consent, which was sent via email and returned signed before participation in the focus group.

For the quantitative phase, a total of 253 individuals participated. All respondents met the inclusion criterion of being researchers who currently conduct or have previously conducted research in Latin America. Recruitment for the quantitative phase took place between June 17, 2025 and July 23, 2025. All participants were adults and participation was voluntary and anonymous. An electronic informed consent form was provided as the first page of the online survey; acceptance of this consent was required to proceed with the questionnaire. The consent form explicitly stated that participation was anonymous, voluntary, and that responses would be used for analysis, publication, and dissemination in open repositories.

In both phases, no personally identifiable information (such as names or email addresses) was collected, and participants could withdraw at any point without completing the study. This study was reviewed and approved by the Responsible Conduct of Research Committee of the Faculty of Psychology at the University of Buenos Aires (approval code: CEI25001).

## Materials and methods

**Focus group.** For the qualitative phase of the study, a discussion guide was collaboratively and iteratively developed by the research team to encourage participants to share their experiences with international funding applications. The development process included multiple rounds of review and refinement to ensure the analytical relevance of the questions and alignment with the study's objectives. This collaborative approach enhanced the credibility of the instrument while fostering critical reflection on the positionalities of the research team.

The final version of the guide was shared with participants in advance, allowing them to reflect on the proposed themes and prepare for a meaningful exchange. It included interactive exercises designed to stimulate engagement and encourage the sharing of personal narratives, while leaving space for new and unexpected themes to emerge organically. The guide can be found here http://doi.org/10.17605/osf.io/fw9nk.

The online focus group was conducted in April 2025. The session, held in Spanish and moderated by team members trained in qualitative methods, lasted approximately 90 minutes. With participants' consent, the session was recorded and subsequently transcribed verbatim for analysis.

**Quantitative survey.** For the quantitative phase, we constructed a self-administered survey based on the themes that emerged during the qualitative phase. We piloted the first version with 5 researchers from the community who suggested modifications and adjustments. The final version included 23 items, distributed across the following thematic sections: sociodemographic and academic background, research trajectory, experience applying to national and international funding calls, strategies for preparing applications, access to non-competitive funds, perceived barriers, and training needs. Questions employed multiple formats, including single- and multiple-choice items, Likert-type scales, open-ended text fields, and validated numeric ranges. Conditional logic was applied so that some questions were only displayed based on previous responses (e.g., only respondents who reported applying to international grants were asked about success rates). The estimated completion time for the survey was approximately 15 minutes. The final version of the questionnaire, data, and analyses script can be found here http://doi.org/10.17605/osf.io/uvdqf. Responses from participants with less than 80% of the questions answered were excluded from data analysis.

The questionnaire was implemented in SurveyMonkey, which allowed for anonymous participation, logic branching, and validation of responses. It was administered in Spanish and circulated between June 17, 2025, and July 23, 2025, through academic networks, institutional mailing lists, and social media platforms.

## Data analysis

Qualitative data were analyzed using the six-phase thematic analysis approach proposed by Braun and Clarke [24] following an inductive and semantic [25] approach. This orientation prioritized the voices and lived experiences of the participants, rather than imposing predefined categories, which was particularly appropriate given the exploratory nature of the study. First, we underwent thorough familiarization with the data, reading and re-reading the full transcript while taking

detailed notes to capture initial impressions, emotionally charged segments, and patterns of meaning. This process laid the groundwork for the second phase—initial coding—where the transcript was examined line by line to identify meaningful units and themes that captured both explicit barriers and more implicit assumptions or tensions expressed by participants.

Initial coding was followed by axial coding, in which codes were grouped into provisional categories based on semantic similarities, causal relationships, or shared narrative structures. The grouping of themes was informed by recurrence, intensity, prevalence [26], and narrative richness, prioritizing dimensions that revealed insights into participants' experiences, recommendations, and ideas within funding ecosystems.

For the survey data, we calculated descriptive statistics. We reported absolute and relative frequencies (n, %) for categorical variables, as well as mean, standard deviation (SD), and 95% confidence intervals (CI) for the mean for numerical variables. All statistical analyses were conducted in R 4.5.1 [27].

This study was preregistered in the Open Science Framework (OSF) prior to data collection, ensuring transparency of the research objectives, methodology, and planned analyses. The preregistration protocol is publicly available at https://osf.io/7b59j.

## Results

### Thematic analysis of the focus group data

The thematic analysis yielded three overarching themes: (1) Strategies and Good Practices for Navigating Funding Applications, (2) Barriers and Structural Constraints Limiting Access to International Funding, and (3) Recommendations for More Inclusive International Funding Practices. While detailed subthemes and illustrative quotations are presented in the S1 Table, several key insights emerge across themes.

Participants described practical approaches—such as early administrative planning, adapting tone and language to reviewers, and leveraging peer review and collaborations—that can improve proposal quality. However, they acknowledged that these strategies often depend on access to supportive networks and resources that are not equally available to all researchers.

Structural issues included opaque national calls, language and cultural barriers, and perceived biases from funders in the Global North. Participants emphasized that some requirements in international calls assume access to infrastructure or credentials that are not universally available, creating unrealistic expectations and excluding otherwise capable researchers. Limited opportunities for international networking and visibility further exacerbate this barrier.

At the same time, participants proposed actionable recommendations to foster fairness and inclusivity in research funding, primarily directed at international funders seeking to increase the representation of Global South researchers in their funding calls. Given their central role in shaping eligibility criteria, evaluation standards, and support mechanisms in cross-border funding schemes, these funders were seen as key actors in enabling the participation of Latin American researchers. The recommendations included expanding calls that explicitly include LMICs, valuing Global South's unique comparative advantages and context-specific knowledge, and providing tailored training opportunities aligned with international funding requirements. Capacity-building initiatives were envisioned as practical, collaborative, and hands-on, with mentorship and simulated review processes highlighted as particularly effective in building familiarity with international standards and expectations.

Collectively, these insights point to context-sensitive approaches that can strengthen capacity, foster participation, and increase the likelihood of success in international funding calls.

### Survey participants characteristics

The final sample included 202 participants who completed at least 80% of the survey. Respondents were based in 16 Latin American countries, with the highest representation from Argentina (n = 126, 62.38%), Mexico (n = 21, 10.40%), Bolivia

(n = 11, 5.45%), and Colombia (n = 10, 4.95%). Most identified as women (n = 143, 70.79%), followed by men (n = 52, 25.74%), and a minority identified as non-binary or preferred not to disclose their gender (n = 7, 3.47%). More than half of participants (n = 104, 51.49%) self-identified as belonging to one or more systematically excluded groups within the scientific community—most frequently, women and gender minorities (n = 81, 77.88%), individuals from socioeconomically disadvantaged backgrounds (n = 19, 18.27%), Indigenous people (n = 12, 11.54%), and persons with disabilities (n = 10, 9.62%).

Most participants held a graduate-level degree, with 59.90% (n = 121) having completed a Ph.D. and 13.86% (n = 28) a master's degree. The sample included researchers from diverse disciplinary fields, including natural and exact sciences (n = 79, 39.11%), health sciences (n = 74, 36.63%), social sciences (n = 53, 26.24%), humanities (n = 21, 10.40%), and engineering (n = 16, 7.92%). The average number of years conducting research was 14.55 (n = 201, SD = 8.59, 95% CI [13.36, 15.75]), ranging from early-career researchers to senior academics. A substantial proportion of participants reported holding a permanent research position (n = 127, 62.87%) and having experience leading research teams (n = 125, 61.88%) or supervising graduate students and postdoctoral researchers (n = 119, 58.91%). A third of respondents (n = 62, 30.69%) had conducted part of their scientific career in HICs, with durations ranging from 1 to 20 years (M = 4.15, SD = 3.38, 95% CI [3.29, 5.00]).

### Experience with funding applications

Regarding funding experience, 80.20% (n = 162) of participants had applied to national calls and 60.40% (n = 122) to international ones. A total of 668 international grant applications were reported by 121 researchers (mean submissions per researcher: M = 5.52, SD = 6.04, 95% CI [4.43, 6.61]). 43.56% (n = 88) of all respondents reported having received international funding at least once. The overall success rate across all applications was 40.60% (M = .41, SD = .36, 95% CI [.34, .47]). The distribution of the participants' success rate over the years of research experience is presented in Fig 1.

When asked about their usual practices in preparing grant applications, the most commonly reported actions (rated "always") included reviewing eligibility and call requirements (n = 151, 82.97%), conducting a detailed reading of the call guidelines and annexes (n = 143, 78.57%), and alignment of the project objectives with the call (n = 129, 70.88%). Less frequently applied were practices such as requesting feedback from colleagues (n = 84, 46.15%), studying the profiles of funder institutions and reviewers (n = 52, 28.57%), and establishing strategic partnerships with other institutions or researchers (n = 35, 27.47%) (Fig 2).

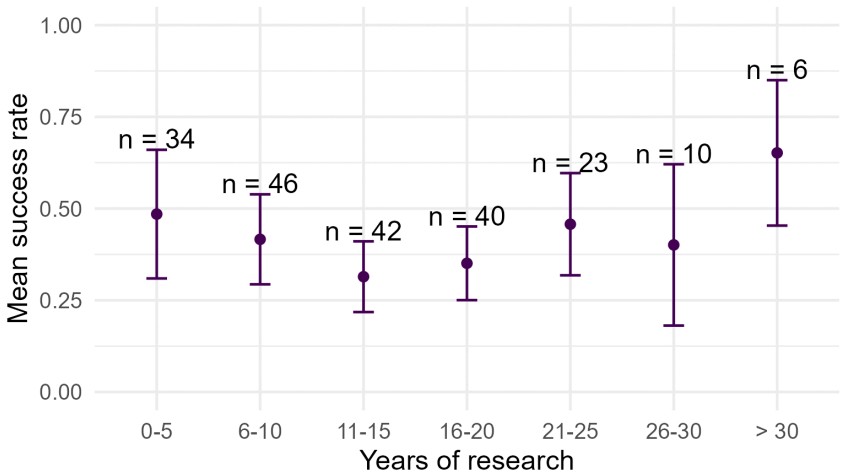

**Fig 1. Distribution of participants' success rate by year of research experience.** Above error bars n represents the number of participants.

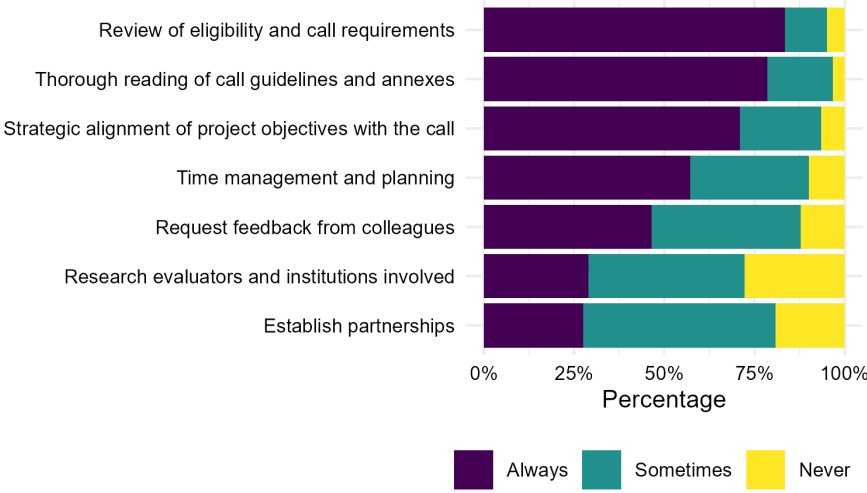

**Fig 2. Reported practices in preparing grant applications.**

## Perceived barriers to accessing international funding

Perceived barriers to international funding were assessed using a set of 10 Likert-scale items (n = 182 complete responses). The vast majority of respondents identified economic barriers to attending international networking events as a major obstacle to accessing international research funding, with more than nine out of ten participants (n = 168, 92.31%) agreeing or strongly agreeing with this statement. The misalignment between the eligibility requirements of international calls and local academic trajectories (e.g., time since Ph.D., position titles) was endorsed by 80.77% (n = 147) of respondents, and rhetorical differences in writing style by 75.27% (n = 137). Psychosocial and perception-related barriers were also salient. Many reported perceiving self-doubt in Latin American researchers regarding their capacity to compete at an international level (n = 126, 69.23%) and, at the same time, not being accustomed to self-promotion practices (n = 124, 68.13%). In addition, 60.44% (n = 110) highlighted biased perceptions from funders about Latin American scientific and infrastructural capabilities as a barrier. Language-related challenges were endorsed by just over half of participants, with 53.85% (n = 98) reporting the use of a non-native language as a barrier (Fig 3).

By contrast, few participants identified issues of clarity in funding information (n = 42, 23.08%) or low regional representation in international networks (n = 26, 14.29%) as critical barriers.

Open-ended responses (n = 47, 25.82%) highlighted additional barriers. Specifically, the misalignment between Global North funders' agendas and local priorities; limited training on how funders operate and make decisions; structural exclusion through elitist or closed research networks; bureaucratic hurdles due to lack of institutional support and difficulties receiving international transfers. Respondents also noted visibility constraints due to limited access to open-access publishing and bibliographic databases, as well as limited familiarity with international evaluation metrics.

## Discussion

Over the past decades, Latin America has made notable advances in scientific productivity, with an increasing number of publications and strengthened research networks. Nevertheless, researchers in the region continue to operate under structural conditions that shape their competitiveness in international funding environments in ways that differ from those of researchers based in HICs [12]. Linguistic, institutional, and contextual factors intersect to influence how Latin American researchers prepare, submit, and experience international grant applications [28].

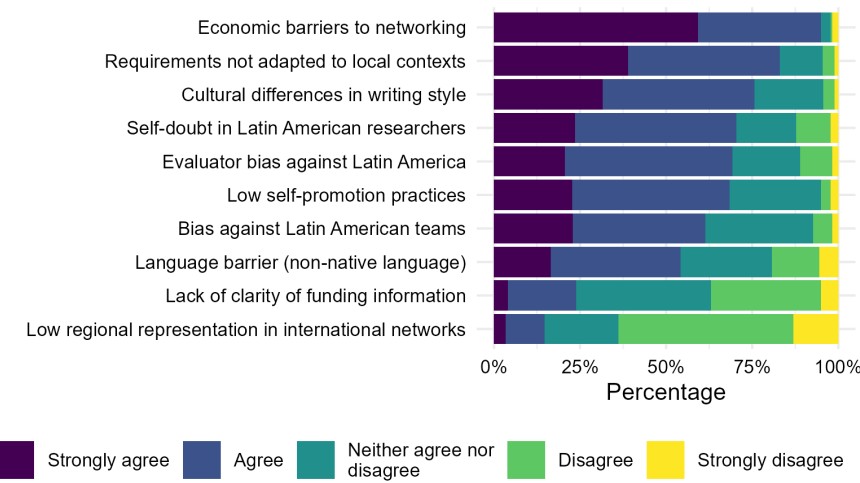

**Fig 3. Perceived barriers to accessing international research funding among Latin American researchers.**

In this study, we use a mixed-methods design to examine barriers to international research funding among Latin American researchers. The focus group qualitative analysis, together with survey responses to closed and open-ended questions, converged on similar structural and procedural challenges. Consistent with previous research [17,29–31], the requirement to write grant proposals in English emerged as a key barrier. Participants described how this linguistic demand not only compromises the clarity and persuasiveness of proposals but also reinforces perceptions of lower competitiveness. Cultural differences in rhetorical conventions further influence proposal development, as international funders often prioritize concise, action-oriented writing styles that may differ from the more expository or theory-driven traditions prevalent in the region [30,31].

Participants also highlighted how features of grant review processes can interact with these linguistic and cultural dynamics. Evaluation criteria and review norms developed within predominantly English-speaking contexts may inadvertently disadvantage researchers from historically underrepresented regions and groups [29,32]. This, together with psychosocial factors such as self-doubt and regional norms that sanction overt self-promotion, was described as shaping proposals where applicants hedge claims, default to passive voice and collective authorship, understate outcomes and novelty, avoid explicit asks regarding budget and resources, and omit quantifiable milestones and track record highlights. Reviewers may read these features as lower confidence, feasibility, and leadership.

These findings underscore that grant writing differs significantly from academic writing and constitutes a specialized professional skill that requires specific training and institutional support [33]. However, the lack of accessible training programs in Spanish or Portuguese that address both linguistic and cultural mismatches further limits Latin American researchers' ability to compete on equal footing with their peers in the Global North. Training or offering support in writing and framing proposals, tailored to international rhetorical expectations while maintaining local voices and their relevance, may help mitigate the disadvantages associated with cultural style differences.

Our analyses also underscored visibility and dissemination constraints that respondents perceived as directly limiting their chances of securing funding. In line with Boudry [34], respondents pointed to paywalled literature and to open-access processing charges (APCs) as structural barriers that disproportionately impact LMIC authors, as they skew who gets read and cited. Respondents also linked limited visibility to difficulties attending networking events, which, in turn, diminishes their access to opportunities. This coincides with Mocanu et al. [35], which concludes that what predicts the success of grant applications is, largely, the PI's previous publications and their international visibility. Enabling travel grants, virtual partnering, and mentorship linking applicants with experienced PIs could mitigate these limitations.

Administrative load and weak institutional support also emerged as another limiting factor. As suggested by previous studies [36–38], substantial administrative burden in grant preparation and management diverts time from research, and this disproportionately impacts less-resourced institutions [39]. Including LMIC experts in the design of funding calls, calibrating expectations about organisational models, and valuing contextual knowledge may reduce the current asymmetries [40].

Beyond linguistic and cultural factors, several structural features of the global research funding landscape continue to shape Latin American researchers' participation in international funding competitions [40]. While international funding schemes are not designed to address structural disparities between research systems, some funding organizations have begun to explore ways of promoting diversity and broader participation within the global research ecosystem. Current and former initiatives by funders such as the Wellcome Trust, the International Development Research Centre (IDRC), the Gates Foundation, and the Chan Zuckerberg Initiative illustrate an emerging interest in equity, diversity, and inclusion (EDI). Although a systematic assessment of such initiatives falls outside the scope of this study, our findings highlight the importance of evidence-informed approaches to understanding how structural conditions shape researchers' experiences. In this context, the development of standardized indicators and transparent data collection practices may help assess the effects of EDI-oriented measures and clarify how different support mechanisms interact with existing funding processes, particularly for researchers based in LMICs [41].

This work has some limitations that should be taken into consideration when interpreting the results. First, the measures rely on self-report; we did not independently verify the number of submissions or awards, nor did we track proposals and career paths longitudinally. As a result, recall bias and social desirability bias cannot be ruled out. Second, the analysis focuses primarily on funding success and perceived barriers in international grant competitions. We did not examine other relevant indicators of research outcomes, such as reviewer scores, subsequent network development, or publication trajectories, which may capture additional benefits associated with applying for international funding. Third, our sample over-represents researchers based in Argentina, reducing regional generalizability. Finally, several challenges discussed in this study—such as limited research infrastructure, institutional capacity constraints, and administrative burdens—are closely linked to domestic science policy contexts. These factors fall in part outside the scope of international funding design and cannot be readily addressed through modifications to international funding schemes alone.

Taken together, the present findings contribute to a more nuanced understanding of how structural conditions shape researchers' competitiveness in international funding environments and where targeted, feasible forms of support may be most effective. Enabling mechanisms informed by the experiences of Latin American researchers, such as those focused on proposal preparation, institutional capacity, and navigation of funding processes, have the potential to strengthen researchers' preparedness for international funding competitions in the near to mid term. Future work should build on these findings and assess the impact of potential interventions.

## Supporting information

**S1 Table. Focus group detailed subthemes and illustrative quotations.**
(DOCX)

## Acknowledgments

We are deeply grateful to all the researchers who generously committed their time to participate in the focus groups and to completing the survey. Their contributions were invaluable to the development of this study. We also sincerely thank the reviewers for their careful reading of the manuscript and for their insightful and constructive feedback, which helped to improve the clarity and quality of this work.

## Author contributions

**Conceptualization:** Jesica Formoso, Paz Miguez, Laura Ación.

**Data curation:** Jesica Formoso, Paz Miguez, Irene Vazano.

**Formal analysis:** Jesica Formoso, Paz Miguez.

**Funding acquisition:** Nicolás Palopoli, Laura Ación.

**Investigation:** Paz Miguez, Irene Vazano, Julián Buede.

**Methodology:** Jesica Formoso, Juan Pablo Barreyro, Laura Ación.

**Project administration:** Laura Ación.

**Software:** Jesica Formoso.

**Supervision:** Laura Ación.

**Visualization:** Jesica Formoso, Julián Buede.

**Writing – original draft:** Jesica Formoso.

**Writing – review & editing:** Nicolás Palopoli, Juan Pablo Barreyro, Laura Ación.

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
