## [Decision Letter · Decision Letter 0]

4 Dec 2025

Dear Dr. Formoso,

Thank you for submitting your manuscript to PLOS ONE. After careful consideration, we feel that it has merit but does not fully meet PLOS ONE’s publication criteria as it currently stands. Therefore, we invite you to submit a revised version of the manuscript that addresses the points raised during the review process.

We look forward to receiving your revised manuscript.

Kind regards,

Claudia Noemi González Brambila, Ph.D.

Academic Editor

PLOS One

Journal Requirements:

2. Please include captions for your Supporting Information files at the end of your manuscript, and update any in-text citations to match accordingly. Please see our Supporting Information guidelines for more information: http://journals.plos.org/plosone/s/supporting-information ..

Reviewers' comments:

Reviewer's Responses to Questions

**Comments to the Author**

1. Is the manuscript technically sound, and do the data support the conclusions?

Reviewer #1: Yes

Reviewer #2: Yes

2. Has the statistical analysis been performed appropriately and rigorously?

Reviewer #1: Yes

Reviewer #2: Yes

3. Have the authors made all data underlying the findings in their manuscript fully available?

Reviewer #1: Yes

Reviewer #2: Yes

4. Is the manuscript presented in an intelligible fashion and written in standard English?

Reviewer #1: Yes

Reviewer #2: Yes

Reviewer #1: It has made a deep reflection on my part, being my experience to grant writing similar to several of the subjects, a very insightful contribution, however I would recomend adding a conclusion and recomendation section if possible after the discussion, so it properly hightlights the important findings that seem obvious but are not.

Excellent work

Reviewer #2: Congratulations to the research team. The topic is essential for Latin American science, and the paper addresses it with rigor and relevance. The mixed-methods approach is well conceived and executed, producing high-quality insights and practical recommendations. The manuscript makes a valuable contribution to understanding the barriers Latin American researchers face in accessing international funding, as ultimately, we all aspire to see a greater number of Latin American researchers securing international funding and contributing more visibly to the global scientific community.

My observations are grouped into two areas: one concerning the contextual framing of the introduction and another addressing the use of terminology around equity and fairness (see details in attached file).

---

## [Author Response · Author response to Decision Letter 1]

9 Jan 2026

Dear Editors of PLOS ONE,

On behalf of my co-authors, I would like to thank you and the reviewers for the careful evaluation of our manuscript, “Perceived barriers for accessing international research funding among Latin American researchers.” We are grateful for the constructive and insightful comments, which have helped us to substantially strengthen the clarity, framing, and contribution of the manuscript.

In response to the reviewers’ suggestions, we have undertaken a thorough revision of the manuscript. The main changes include: (i) a substantial reframing and expansion of the Introduction to situate the study within the competitive and globalized landscape of contemporary science; (ii) a more precise use of terminology related to equity, fairness, and asymmetry in international funding; (iii) clearer identification of the intended audience for the recommendations; and (iv) the addition of a strengthened concluding section integrated into the Discussion to explicitly highlight key findings, implications, and future directions.

Below, we respond to each reviewer's comment point by point. Reviewer comments are reproduced in italics, followed by our responses. All changes are reflected in the revised manuscript unless otherwise noted.

Reviewer 1

It has made a deep reflection on my part, being my experience to grant writing similar to several of the subjects, a very insightful contribution, however I would recommend adding a conclusion and recommendation section if possible after the discussion, so it properly highlights the important findings that seem obvious but are not.

Response:

Thank you for this positive and thoughtful feedback.

We agree that adding a clearer conclusion and recommendations helps highlight the main findings and their implications. Since adding an entire section after the discussion would not align with the journal's structure, we expanded the final paragraphs of the Discussion to serve as a succinct but clear concluding synthesis (page 15, paragraph 3).

In addition, we revised all the Discussion section and it now explicitly highlights: (i) where structural asymmetries arise across the funding pipeline, (ii) which barriers are most amenable to targeted intervention, and (iii) how context-sensitive supports informed by researchers’ lived experiences may translate into near- to mid-term improvements in competitiveness. We also clarified directions for future research to evaluate the impact of such interventions.

Reviewer 2

General Comment

Congratulations to the research team. The topic is essential for Latin American science, and the paper addresses it with rigor and relevance. The mixed-methods approach is well conceived and executed, producing high-quality insights and practical recommendations. The manuscript makes a valuable contribution to understanding the barriers Latin American researchers face in accessing international funding, as ultimately, we all aspire to see a greater number of Latin American researchers securing international funding and contributing more visibly to the global scientific community.

Response: Thank you very much for your generous and encouraging assessment of our manuscript. We are particularly grateful for the detailed conceptual suggestions, which prompted substantial improvements in the study's framing, terminology, and positioning.

For clarity, my observations are grouped into two areas: one concerning the contextual framing of the introduction and another addressing the use of terminology around equity and fairness.

Area 1: Contextualization of the Introduction

The paper would benefit from opening with a broader overview of how science operates in the 21st century, to situate the reader before narrowing to the Latin American perspective. Moving from a frame of “this is an inequity that must be corrected” to “this is the competitive landscape of global science, and here is how Latin American researchers are structurally disadvantaged within it” would strengthen the rationale for the study and clarify the interpretation of its results.

Response: We fully agree with this suggestion and have substantially revised the Introduction's opening to provide a broader contextual framing of global science as a competitive, internationalized enterprise. The revised Introduction now begins by situating research funding within global collaboration networks, standardized evaluation regimes, and increasing competition for limited resources, before narrowing to the specific structural position of Latin American researchers. This reframing explicitly shifts the narrative from one centered on moral inequity to one grounded in structural disadvantage within a competitive global system, thereby strengthening the interpretability of the findings and anchoring the study in the realities of contemporary science. These changes are reflected in the first two paragraphs of the Introduction.

It could cover aspects such as the following:

1. Science as a globalized enterprise that largely operates in English and functions through international knowledge networks.

Response:

i. We added an opening paragraph describing the current science ecosystem:

“Scientific research has become a highly globalized enterprise, increasingly organized through international collaboration networks, competitive funding schemes, and standardized evaluation criteria. While global integration has expanded opportunities for knowledge circulation, it has also reinforced long-standing structural imbalances in research capacity across countries and regions (1). Researchers based in high-income countries typically have access to more robust funding, infrastructure, institutional backing, and international research networks, whereas researchers in low-to-middle-income countries (LMICs) encounter persistent structural constraints associated with comparatively fragile and under-resourced national scientific systems (2,3). As the number of researchers competing for limited funding continues to grow faster than overall research budgets, these pre-existing differences increasingly translate into asymmetric research capacities and highly uneven scientific outputs (4). ”.

ii. We described in more detail the predominance of English as the main language of scientific communication.

2. Researchers from wealthier countries benefit from greater resources, infrastructure, institutional support, and network participation, while those in Latin America face multiple structural limitations stemming from weaker and less well-funded national science systems.

Response: We incorporated this point into the opening paragraph and it is described in more detail when developing specific aspects in which LMICs and HMICs researchers’ environments differ throughout the manuscript.

3. Public R&D funding is designed to serve national goals (economic competitiveness, innovation, social development, etc.), and international R&D funding, when it involves public money, is driven by strategic (not equity) objectives.

Response:

i. We appreciate this clarification and have revised the complete manuscript accordingly. We incorporated this notion explicitly on page 3, paragraph 2: “Public investment in research and development is primarily oriented toward advancing domestic priorities such as economic competitiveness, innovation, and social development, while international R&D funding that relies on public resources is often guided by strategic considerations rather than by redistributive objectives (9,10)”

4. The number of researchers competing for limited funds continues to grow faster than the volume of available resources; access to international funding will therefore become increasingly competitive.

Response: We incorporated this point into the opening paragraph.

5. Grant writing is a specialized skill, one that even researchers in the Global North must deliberately cultivate, supported by professional structures developed for this purpose.

Response:

We integrated this point explicitly into the Introduction and highlighted what was already present in the Discussion section.

i. Introduction: “Further more, grant writing itself constitutes a specialized professional skill that requires deliberate training, familiarity with funders’ expectations, and institutional support. In many higher-income countries, researchers operate within well-established support structures—such as research development offices, grant administrators, internal review processes, mentorship, and targeted training—that enhance preparedness for highly competitive funding calls (21). By contrast, researchers in less well-resourced scientific systems often have limited access to such support, with proposal development frequently absorbed into existing academic workloads.” (page 4, paragraph 2).

ii. Discussion: “These findings underscore that grant writing differs significantly from academic writing and constitutes a specialized professional skill that requires specific training and institutional support (33).” (page 13, paragraph 3).

Area 2: Use of Terminology and Framing of Equity and Fairness

The use of terms such as “financial inequity”, “equitable access”, and “fairness” should be reconsidered for precision and realism. International funding is not, and has never been, primarily an instrument for global redistribution or equity. Framing it as such can weaken the argument. The concepts of asymmetry, disparity, or imbalance might better capture the dynamics at play without implying moral obligation or entitlement.

Response: We appreciate this important conceptual guidance and have carefully revised our terminology throughout the manuscript. We reduced normative language related to “equity” and “fairness” and replaced it, where appropriate, with terms such as structural asymmetries, disparities, and competitiveness imbalance. Where the term “equity” is retained, it is now clearly framed as an aspirational or emerging objective rather than as an implicit mandate of international funding systems.

Detailed comments:

1. Access to international research funding is merit-based, not equity-based; calls are not designed to ensure equal opportunity across countries, institutions, or researchers, but to support the most competitive proposals. The issue is therefore not the distribution of resources but rather how structural imbalances in national systems affect researchers’ competitiveness.

Response: We fully agree and have revised the manuscript to explicitly reflect this distinction. The revised text clarifies that international funding calls are designed to support the most competitive proposals and that the core issue examined in this study is how structural conditions within national research systems shape researchers’ ability to compete, rather than the distributional logic of funding itself.

2. The authors could highlight that some funders are beginning to recognize their potential role in promoting a more equitable and diverse global research ecosystem (for example, the Chan Zuckerberg Initiative, National Geographic Society). This would allow the paper to frame “equity” as an emerging and desirable aspiration for progressive funders, rather than as a fundamental failure of the current system.

Response: Thank you for this valuable suggestion. We now explicitly include this point in the Discussion: “While international funding schemes are not designed to address structural disparities between research systems, some funding organizations have begun to explore ways of promoting diversity and broader participation within the global research ecosystem. Current and former initiatives by funders such as the Wellcome Trust, the International Development Research Centre (IDRC), the Gates Foundation, and the Chan Zuckerberg Initiative illustrate an emerging interest in equity, diversity, and inclusion (EDI). Although a systematic assessment of such initiatives falls outside the scope of this study, our findings highlight the importance of evidence-informed approaches to understanding how structural conditions shape researchers’ experiences. In this context, the development of standardized indicators and transparent data collection practices may help assess the effects of EDI-oriented measures and clarify how different support mechanisms interact with existing funding processes, particularly for researchers based in LMICs (41).“ (Page 14, paragraph 3).

3. Several challenges described in the paper, such as limited infrastructure or institutional weakness and red tape, stem from domestic science policy gaps that cannot realistically be solved through modifications in international funding design.

Response: We included this point explicitly in the Discussion section as a limitation of the results: “Finally, several challenges discussed in this study—such as limited research infrastructure, institutional capacity constraints, and administrative burdens—are closely linked to domestic science policy contexts. These factors fall in part outside the scope of international funding design and cannot be readily addressed through modifications to international funding schemes alone.” (page 15, paragraph 1)

4. The expression “misalignment between donor priorities and local challenges” should be revised; “differences between donor priorities and local challenges” would be more accurate, as donor priorities are defined according to their own strategic interests.

Response: We changed the word “misalignment” for “differences” (Page 5, line 8).

5. When referring to “recommendations for more equitable funding practices”, clarify who the intended audience is (international funders, national agencies, or research institutions?) since each operates with different mandates and responsibilities.

Response: We have addressed this comment by specifying the intended audience of the recommendations. The revised Results and Discussion now state that the recommendations emerging from the qualitative data are primarily directed at international funders seeking to broaden participation and representation of researchers from LMICs in cross-border funding schemes:

i. “At the same time, participants proposed actionable recommendations to foster fairness and inclusivity in research funding, primarily directed at international funders seeking to increase the representation of Global South researchers in their funding calls. Given their central role in shaping eligibility criteria, evaluation standards, and support mechanisms in cross-border funding schemes, these funders were seen as key actors in enabling more equitable participation.” (Page 9, paragraph 5)

6. If the authors wish to emphasize the importance of diversity in science (which it is), they could suggest local and international mechanisms to promote it (e.g., mentorship schemes, scholarships for underrepresented groups, etc) but I think this was not the focus of the study.

Response: We agree that diversity is a critical issue; however, as suggested by the reviewer, we have kept it secondary to the main focus of the study to ensure conceptual coherence and alignment with the study’s empirical scope.

Based on the modifications made to the manuscript, we removed the following references:

• Kozlowski D. Topics and institutions in the reproduction of intersectional inequalities in science [PhD thesis]. Luxembourg: Université du Luxembourg; 2023.

• Jebsen JM, Abbott C, Oliver R, Ochu E, Jayasinghe I, Gauchotte-Lindsay C. Review of barriers women face in research funding processes in the UK. Psychol Women Equal Sect Rev. 2020;3(1–2):3–14.

• Torres IL, Collins RN, Hertz A, Liukkonen M. Policy proposals to promote inclusion of caregivers in the research funding system. Front Educ. 2024;9.

• Narayan A, Chogtu B, Janodia M, Radhakrishnan R, Venkata SK. A bibliometric analysis of publication output in selected South American countries. F1000Res. 2023;12:1239.

At the same time, we included the following references:

• Chinchilla-Rodríguez Z, Miao L, Murray D, Robinson-García N, Costas R, Sugimoto CR. A global comparison of scientific mobility and collaboration according to national scientific capacities. Front Res Metr Anal. 2018;3:17.

• Murray M, Mubiligi J. An approach to building equitabl

---

## [Decision Letter · Decision Letter 1]

10 Feb 2026

Perceived barriers for accessing international research funding among Latin American researchers

PONE-D-25-50701R1

Dear Dr. Formoso,

We’re pleased to inform you that your manuscript has been judged scientifically suitable for publication and will be formally accepted for publication once it meets all outstanding technical requirements.

Kind regards,

Claudia Noemi González Brambila, Ph.D.

Academic Editor

PLOS One

Additional Editor Comments (optional):

Reviewers' comments:

Reviewer's Responses to Questions

**Comments to the Author**

Reviewer #2: All comments have been addressed

Reviewer #3: All comments have been addressed

2. Is the manuscript technically sound, and do the data support the conclusions?

Reviewer #2: Yes

Reviewer #3: Yes

3. Has the statistical analysis been performed appropriately and rigorously?

Reviewer #2: Yes

Reviewer #3: Yes

4. Have the authors made all data underlying the findings in their manuscript fully available?

Reviewer #2: Yes

Reviewer #3: Yes

5. Is the manuscript presented in an intelligible fashion and written in standard English?

Reviewer #2: Yes

Reviewer #3: Yes

Reviewer #2: The authors clearly addressed all my comments and the new version benefits from a more comprehensive contextualization and clarity of the issue discussed in the paper. It does make the case for the need to support researchers from Low and Middle Income Countries (LMIC) to increase access to international research funding.

Reviewer #3: Solid improvements have been made to the revised draft. Well one. I have no further comments from my end.

---

## [Editor Report · Acceptance letter]

PONE-D-25-50701R1

PLOS One

Dear Dr. Formoso,

I'm pleased to inform you that your manuscript has been deemed suitable for publication in PLOS One. Congratulations! Your manuscript is now being handed over to our production team.

Kind regards,

on behalf of

Dr. Claudia Noemi González Brambila

Academic Editor

PLOS One